# Talent Management in the Banking Sector: A Systematic Literature Review

Unnar Theodorsson *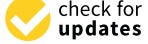, Thorhallur Gudlaugsson and Svala Gudmundsdottir

Department of Business Administration, School of Social Sciences, Gimli Sæmundargata, University of Iceland, 102 Reykjavík, Iceland; th@hi.is (T.G.); svala@hi.is (S.G.)
* Correspondence: uft1@hi.is

**Abstract:** This paper conducts a systematic literature review and relates to how talent management (TM) and recruitment strategies are applied in the financial sector, and specifically in the banking sector. The goals of this paper are to provide a comprehensive analysis of TM research in the financial sector, identify and debate major research topics, and suggest future research possibilities. The study examines publications that were published between 2000 and 2021, which were obtained from the databases Web of Science and Scopus. According to the findings, academics are becoming more interested in TM in the banking sector, which appears to be linked to the severe scarcity of skilled people who possess required talents. The findings are relevant to both academia and the banking sector, as the paper provides data relating to what has already been researched in academia, while also highlighting the need for more research into how TM is understood, valued and implemented. Consequently, this paper paves the way for academics to conduct empirical research on TM and recruitment tactics in the banking sector and the financial sector as a whole. This paper is structured according to the PRISMA requirements for systematic literature reviews.

**Keywords:** talent management; human resource management; recruitment strategy; banking industry; financial sector; strategy; SHRM

## 1. Introduction

Adversities, such as the 2008–2009 global financial crisis, emphasize the need for the financial sector to identify and address the significant practices that cause such effects. How knowledge-intensive service businesses are moving from an information-based (Industry 4.0) activity sphere to one that is adapting the principles of a more complex body of operation (Industry 5.0). Adapting the Internet of things, robotics, artificial intelligence, and big data has been a focal point of debate in recent years. Examples of sectors that are involved in this development are the energy sector, the pharmaceutical sector, and the financial sector. Academic candidates (BS, MS, and Ph.D. levels) with stronger specialization skills are required in sectors that are moving towards this way of conducting business. Recruitment processes for organizations in the abovementioned sectors are, therefore, facing a new set of challenges, such as a lack of talent and suitable academics, as well as fiercer competition, both from within the banking sector and from other sectors. Technology- and data-related positions, such as those of data scientists and quantitative analysts, are among the most in demand, which implies that large banks, for example, will face renewed competition for talent (Min 2018) The need for banks to develop dependable techniques to recruit competent managers and ensure a high retention of important workers in the future has grown. Human resource management (HRM) is one of the most important components in the financial industry for lucrative and efficient banking operations, the development of new banking products, and the provision of improved services to consumers (Haines and St-Onge 2012; Vemic-Djurkovic et al. 2013).

The need for managerial characters is more important than ever, since the recruiting and onboarding of new staff is a difficult and expensive process, with the cost of losing one

person ranging from tens of thousands of dollars to 1.5 times their yearly wage (Bersin 2013). Talent management (TM) was initially established around the turn of the millennium by McKinsey & Company as a sub-section or new hybrid of HRM (Chambers et al. 1998). TM is claimed to give a business a competitive advantage by allowing it to discover, develop, and redeploy workers with exceptional abilities in a disciplined manner. The retention and commitment of important personnel are intimately linked to TM (Tej et al. 2021).

It is crucial to focus on the difficulties and possibilities that exist within TM through a comprehensive literature study of TM in the financial industry. It is important to conduct a clear and thorough assessment of the available evidence on the subject, both in the financial sector and for academia, so that research into the topic can develop. It is also important to fill the research gaps that have been identified in our current understanding of a field. Such studies and assessments can draw attention to methodological flaws in research projects and be utilized to improve future studies in the field.

The purpose of this study is to perform a systematic evaluation of peer-reviewed, published research on TM concerns, specifically in the banking sector. In doing so, the study captures the scope of research on the issue of TM, identifies major topics and controversies in this literature, and indicates hurdles to growth in the area. Furthermore, we suggest some of the most important research directions for the future. The study delves into the methodologies that have been employed to date in researching TM, as well as theories and methods that provide the best potential for strengthening the field's theoretical and empirical underpinnings.

With the goal of identifying how the financial industry incorporates TM into its activities, a comprehensive literature study is conducted to map the current information about how TM is applied in the recruitment tactics of banks and the financial sector. The findings are relevant to both academia and the financial sector, as the paper provides evidence on previous academic studies, while also highlighting the need for additional research on financial industry practices in terms of how TM impacts employee job satisfaction. The paper includes the following elements:

1.  A review of how TM has been conceptualized and developed in the literature;
2.  An overview of how TM has been applied in the context of the financial sector and banking sector in the literature;
3.  An identification and discussion of key research topics;
4.  Recommendation avenues for future research.

*Talent Management: A Perspective*

The American consultancy firm McKinsey & Company introduced TM in the article "The war for talent", which was published in 1998 (Chambers et al. 1998). It is important to recognize the challenge that much of the previous literature on the topic TM has been published in the popular literature, rather than in the scientific peer-reviewed literature, which poses a challenge (Iles et al. 2010; Vaiman et al. 2012). Several scientific publications have been published, but such publications are few and far between, which highlights the present schism between practical and scholarly interest in the subject (Dries 2013; Al Ariss 2014) and underlines the necessity for an extended academic focus on the concept of TM. Owing to the lack of a clear and consistent definition of "talent management", the topic's analysis in the scientific literature is in its infancy stage (Lewis and Heckman 2006). A conceptual framework based on empirical research will enable the opportunities for application areas for further research. Collings and Mellahi (2009) confirmed this point.

An advance was noted in the development of TM two years later (Collings et al. 2011). This progress was primarily attributed to U.S. scholars, who had made notable contributions by applying the North American way of thinking and conducting research. The emphasis on TM has also persisted in the business sector, with worldwide corporations leading the way (Powell and Lubitsh 2007). Based on the notion that the objective of applying TM methods is to recruit, develop, motivate, and retain talent (Mccauley and Wakefield 2006; Ready and Conger 2007; Christensen Hughes and Rog 2008; Beechler and

Woodward 2009; Davies and Davies 2010; Meyers and van Woerkom 2014), some academics believe that TM only occurs within an organization. According to Tansley (2011), TM can, thus, be handled in a variety of ways, which are as follows: (1) TM is not employed at all in HRM policy; (2) only some levels of an organization have a common knowledge of TM; and (3) TM is understood and extensively applied in an organization's performance.

Many human resources (HR) professionals and business leaders have witnessed the influence of employing people who are mediocre simply to fill open positions, which became clear in the years of the global economic growth (2002–2007), and an understanding of the significance of hiring the 'right' people has now become a priority for many companies (Dewhurst et al. 2012).

The emphasis on TM implies moving away from more typical HR to strategic TM, which is decided by a business strategy and other activities (Silzer and Church 2009). Parallels between TM and HR can be noted. HR is a management technique that focuses on acquiring talent, onboarding new workers, and transforming these workers into valuable performers for a firm. HR has a longer history that dates back to the early 1900s (Taylor 1914, 2003; Fayol 1949), while TM is a recent concept that takes a more comprehensive approach. TM moves the emphasis from what the firm can extract from the individual to how the company can engage and keep a valuable employee so that they can benefit the company in the long run. In the context of corporate goals, there is a consistent emphasis on involving those closest to the employees in the context of the corporate goals. Top performers are groomed for strategic leadership roles using TM methods. Consequently, instead of token employee appreciation gestures and exit interviews, a greater focus is placed on trainings, personal development, coaching, and stay interviews (Chambers et al. 1998).

According to Collings and Mellahi (2009) the influence of TM significantly impacts an organization's performance. The emphasis is on how, as a consequence of commitment and motivation, effective TM is responsible for an indirect positive influence on the organization's activities (Collings and Mellahi 2009; Hoglund 2012).

Christensen Christensen Hughes and Rog (2008) emphasize that an indirect goal of TM is to create greater employee commitment. When evaluating the employee commitment according to parameters such as better productivity, more successful retention of employees or greater customer satisfaction, Odierno (2015) pointed out that abundantly committed employees achieve better results than employees who are not as committed. Macro factors, such as inter alia, mobility, legislation and internationalization, also profoundly impact on an organization's activities and ultimately its ability to perform in a competitive climate, which leads to the need to go beyond the organizational walls and look further.

Ulrich (2007) illustrated how TM is linked to community outcomes and underlined how a social image can help a business to recruit more competent personnel (Ulrich 2007; Phillips and Roper 2009; Stahl et al. 2012; Egerová 2014).

Combined with the fact that their scopes and intentions are lacking, it should be noted that the terms "talent" and "talent management" are not defined uniformly. Simultaneously, the literature shows a discrepancy in whether or not TM concerns all the employees or solely focuses on high-performing employees or high-potential employees.

Meyers and van Woerkom (2014) stated that there is a clear distinction between the concepts of exclusive and inclusive TM strategies that can be observed in the literature, which applies to the contributions made by both practitioners and academics. Initially, TM was focused on an exclusive strategy that targeted the 1–15% of employees who were deemed as valuable and considered to possess exceptional characteristics (Lepak and Snell 1999), the employees who demonstrated efficient performance and showed a high potential (Silzer and Church 2009) or those who held positions of critical strategic importance (Huselid et al. 2005).

Cappelli (2008) pointed out that a lack of talent forces organizations to employ more forceful tactics when searching for, attracting and selecting the most desirable candidates. It is difficult to attract such candidates, since they are typically more trained and have more opportunities to advance, better salary packages and more overall benefits than other

employees. This difficulty results from most organizations' exclusive TM strategy practices to retain such employees.

Using the argument that high-potential employees, ceteris paribus, add more value to the organization than the employees that are considered average (Aguinis and O'Boyle 2014), it makes sense that high-potential employees' commitment and motivation becomes a priority when prioritizing and allocating the HR budget, assuming that the assets pay off.

When an organization engages in the implementation of an exclusive TM strategy, it is crucial to operate with transparency. A lack of transparency could have a negative effect on other employees, as such employees could end up with false expectations, and if their expectations are not met, these employees could become demotivated. Dries and De Gieter (2014) explained how the latter risk is particularly high in certain organizations (e.g., organizations that do not operate with transparency but with privacy practices in regard to how to approach TM and, hence, do not clearly identify who is included in and who is excluded from the talent pool). In essence, the exclusive TM strategy addresses how an organization can attract candidates who can secure the organization's success in the competitive business environment that the organization is in, despite a limited pool of talent within the labor market. With the objective of stimulating motivation and loyalty, an organization invests in the recruitment and development of a small group of employees. Collings and Mellahi (2009) stated that when it comes to gaining a competitive advantage over competitors, securing the best employees can potentially result in a long-term strategic position of strength.

An exclusive TM approach would be an ideal fit for organizations that are governed by a competitive corporate culture, where employees thrive on incentives for exceptional performance (Meyers and van Woerkom 2014). In addition to exclusive TM, inclusive TM can be used. In contrast to exclusive TM, the inclusive TM method assumes that each employee of an organization has talents that are important to the firm's success. This concept is one of the basic pillars of positive psychology, which emphasizes the objectives in life that are good or operate correctly, according to Seligman and Csikszentmihalyi (2000).

Swailes et al. (2014) defined the inclusive TM approach in the following terms: "The realization that all workers have talent, together with continual evaluation and employment in positions that are most suited and give the most potential (via participation) to employees who possess these skills." The tools that are considered as the major components of this strategy are training and experience gain (McCall 1998). The goal of an inclusive TM approach is to focus on all the workers' talents and provide them with the opportunity to use their potential at work (Meyers 2015). Furthermore, personal development resources are spread over various types of talents. It is relevant to keep in mind that employees could possess talents that are not of value to the organization when an inclusive TM strategy is implemented.

To ensure that talent is not wasted, it is important that in those situations that the organization is aware and facilitates the finding of a more suitable position for the employee (Swailes et al. 2014). According to Meyers (2015), an "inclusive personnel management strategy supports employee well-being, learning, and activity by providing employees with the chance to fully achieve their potential". It is possible to conclude that this strategy allows organizations to respond efficiently to labor market challenges, since the strategy enables them to acquire the best talent in the face of a common talent scarcity, attract a workforce that is more diverse, and maintain a structure that has some agility in terms of adapting to volatile labor markets through investment in various types of talents.

One of the more recent TM areas to attract attention is how TM can be used as a tool in risk management. According to Basco et al. (2021), the firm's level of risk aversion mediates the association between family-owned enterprises and investment in TM methods. Another focus of the recent TM literature is digitalization. To understand each team member's attitude, Vatousios and Happonen (2022) have developed a digital qualitative approach for talent profiling, whose purpose is to recruit the ideal group of candidates and determine ways to improve the retention rate once the team has been assembled. Inclusion, corporate

responsibility, fairness, and equal employment opportunities are the core underlying concepts of a responsible people management strategy. This type of strategy is a strong fit for firms that have a strategic interest in promoting health and well-being (Anlesinya and Amponsah-Tawiah 2020).

Both strategies have pros and cons. A strategy that works flawlessly for one business may not work for another. Therefore, it is essential for an organization to evaluate its organizational foundation and profile when determining which type of TM strategy is the correct fit for it. Factors such as the organization's size, industry, demographics, culture, values, mission, vision, and overall business strategy should all be evaluated in the process. When such factors become clear, a further assessment of the strengths and weaknesses of each strategy is appropriate (Sidani and Ariss 2014; Thunnissen 2016).

More than 24 years have passed since TM was introduced, and no systematic literature review about the financial sector has been conducted. As can be observed, there are many reasons why such a review is necessary. Employees do not necessarily remain in the same organization for long. Therefore, companies must be agile, proactive and attractive. It is important to see TM as a supportive, additional feature that allows the organization to strengthen its position and HRM processes, rather than a substitute feature. Understanding inclusive and exclusive TM is crucial, as TM can be a comprehensive strategy with a broader sustainable structure. The key is to find the right match for an organization in relation to its goals, work environment and profile.

It is important to ensure that high-potential employees can achieve significant results. In this context, one can remember what the costs are. The possible income and benefits are related to the costs and compromises that the organization accepts. Recruitment and onboarding are one issue, but the signals that one sends to the rest of the workforce also play an important role. One must ensure that the initiative is carried out consciously and that the process is transparent. Therefore, it is crucial that a review is conducted so that academics, professionals and other stakeholders gain an insight into and can continue the process of TM in the banking sector and other competence-heavy service industries. Tying the practical and academic by showcasing the gaps, it is necessary to gain an overview of what talent an organization possesses, what the organization requires, and what strategy the organization has the means to create to attract, develop and retain the "right" employees.

## 2. Methods

This paper follows the structure presented by Denyer and Tranfield (2009) for conducting a structured literature review, as was presented in the Handbook of Organizational Research Methods. The systematic literature review is a valuable tool to evaluate the landscape of academic literature concerning a given subject, laying the foundation for academic investigations. Furthermore, the adoption of a systematic procedure for choosing papers for the review reduces bias (Xiao and Watson 2019; Denyer and Tranfield 2009; Tranfield et al. 2003). As mentioned, the goal of conducting a comprehensive literature study for this research is to identify knowledge gaps in the financial industry-related TM and recruitment procedures in order to develop a trustworthy knowledge base. Academics, corporations, and legislators are among those who would benefit from such a study (see Figure 1).

According to Jesson et al. (2011), a systematic literature review "provides a systematic and transparent technique of acquiring, summarizing, and rating the findings of research on a certain topic or question." Consequently, this comprehensive literature review lays the groundwork for empirical investigations by concentrating, inter alia, on TM and how it is applied in the financial industry (see Figure 1).

The study is conducted by gathering information on TM and the financial industry to establish what is already known about the issue and which theories have been used. In addition, the study aims to analyze various concepts in this particular field. To conduct the systematic literature review, as presented by Tranfield et al. (2003), this paper will follow three stages, as shown in Figure 1.

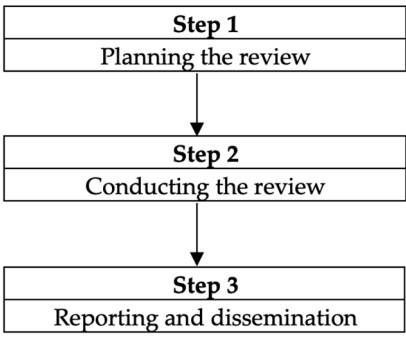

**Figure 1.** The three stages of the review.

*2.1. Planning Review*

As can be observed in Figure 1, the planning stage is the first phase. Here, the proposal is identified and prepared. Furthermore, this is the stage where the need and the protocol for the review are developed (Tranfield et al. 2003). The requirement is to investigate TM and determine how it is handled in the financial industry (Charan 2011). Articles published in peer-reviewed academic journals are the focus of this review. The review protocol excludes other sources, such as books and conference papers, whether peer-reviewed and non-peer reviewed.

Two well-known databases, Web of Science and Scopus, were used to search for scholarly publications. The two databases are among the most comprehensive abstract and citation databases of peer-reviewed literature available and cover a significant number of relevant published articles. The inclusion requirement used for selecting data were papers published in peer-reviewed journals, documents, as well as institutional reports. All the included documents needed to be in English. The systematic literature review method is characterized by the modification of categorizations and classifications and is, thus, inductive (Schreier 2017). Denyer and Tranfield (2009) stated that a transparent structure of selection criteria is required when conducting the selection process and is a necessity when determining whether or not it is appropriate for the study objectives.

The second step involved finding, selecting, and analyzing data quality, as well as the extraction, monitoring, and synthesis procedures (Tranfield et al. 2003).

*2.2. Conducting the Review*

The review's search string was created with the goal of identifying acceptable and relevant peer-reviewed academic publications that meet the review's requirements. The literature was sought using the databases Web of Science and Scopus, as well as the tools that they provide. The standard Boolean terms "AND, OR, and NOT" were applied, and they allowed for the construction of a single search algorithm that ran the selected keywords, and thereby scanning titles and abstracts.

The string-searching algorithm used in the search included the terms "bank", "financial sector", "recruiting strategy", and "talent management"; it was additionally segmented based on relevant topics (see Table 1).

**Table 1.** Keywords and sub keywords that were used as a search string in this systematic literature review's research.

| Keywords | Sub-Keywords |
| --- | --- |
| Recruitment Strategy | Recruitment Strategies |
| Financial Sector | |
| Talent Management | |
| Bank | Banks |
| | Banking |

The final keyword search strings were "bank* or financial sector" and "talent management or recruitment strategy*." The search covered the period from January 2000 to December 2021, and the goal was to cover as much research that was relevant to the issue as possible within this time period. The search returned 197 publications. In January 2022, the search and analysis processes were completed.

Inclusion and Exclusion Criteria

The initial search returned 197 publications. Ninety-seven papers were obtained from Web of Science, and one hundred papers were obtained from Scopus (see Figure 1). With access to the complete scope of the discovered literature, the entire work was read in full by the authors. Using the Jesson et al. (2011) proposed systematic literature review technique, the inclusion selection process involved screening for titles, abstracts, and keywords in each of the 197 sourced publications to determine whether or not a paper should be included in the review. The writers examined the entirety of the articles in a number of situations to ensure that the articles' inclusion was determined accurately.

This approach resulted in the exclusion of 121 articles because they were either irrelevant to the business of banking, TM, and recruitment techniques, or they were not published in English. Following the first screening, 76 publications were studied further in relation to the themes under consideration. This study resulted in the exclusion of 29 articles. Forty-seven publications ultimately met the study's aim.

*2.3. Reporting and Dissemination*

The last stage of the systematic literature review process involves recommendations and reporting (Denyer and Tranfield 2009) to ensure that the evidence found can be activated in a practical context. The qualitative content analysis method was applied in the analytical stage.

By using Microsoft Excel 365, it was possible to convert qualitative data to numerical data in a systematic way (Collis and Hussey 2014). As the frame was paired with data-driven inductive logic for open coding, the 47 selected papers were thoroughly read. Subsequently, it was necessary to find concepts and develop them. By defining and unifying each area, the option for execution was created (Corbin et al. 2015). The frame's foundation was called "categories based on definitions in the publications" and grouped based on search data. It was essential that the researcher could traverse, back and forth, the pool of collected data from the search in order to examine it continually while finishing the frame, which the qualitative content analysis approach combined with open code enabled (Schreier 2017). Accordingly, the foundation was built on a "strategy for discovering concepts in the researcher's data" (Schreier 2017). Hence, this strategy fit the idea of this review.

The coding frame included, but was not limited to, publishing, years, markets, techniques, theories, aims, purposes, and objectives, themes, keywords, and future research recommendations. To ensure that the key topics of each paper were included, these were divided into themes and categories. These categories were "talent management," "recruitment strategies," "banks," and "the financial sector." In the findings section, a further presentation of the outcomes is reflected upon.

## 3. Results

The presentation of the findings of this chapter is arranged as follows: (1) years of publication; (2) research focus by regions and geographical distribution of areas of study; (3) studies by theoretical approach; (4) overview of studies by aim, purpose, and objective; (5) overviews of keywords by industry and frequency of keywords; (6) analysis of studies by key concepts; (7) talent management; (8) recruitment strategies; and (9) contributions and suggestions for future research.

*3.1. Journal and Year of Publication*

The findings indicate a growing interest in TM, recruitment tactics, and the financial sector and the business of banking (see Figure 2). A total of 72% (131 publications) of the 47 studies considered in this analysis were published within a 6-year period from 2015 to 2021. Furthermore, 18% (32 publications) were published from 2010 to 2014, whereas 10% (18 papers) were published between 2000 and 2009. It is worth noting that publications on the themes first emerged in 2001, and that only six papers were published between 2001 and 2004. Furthermore, it should be emphasized that papers published in 2021 comprised everything published to the end of the search in December 2021.

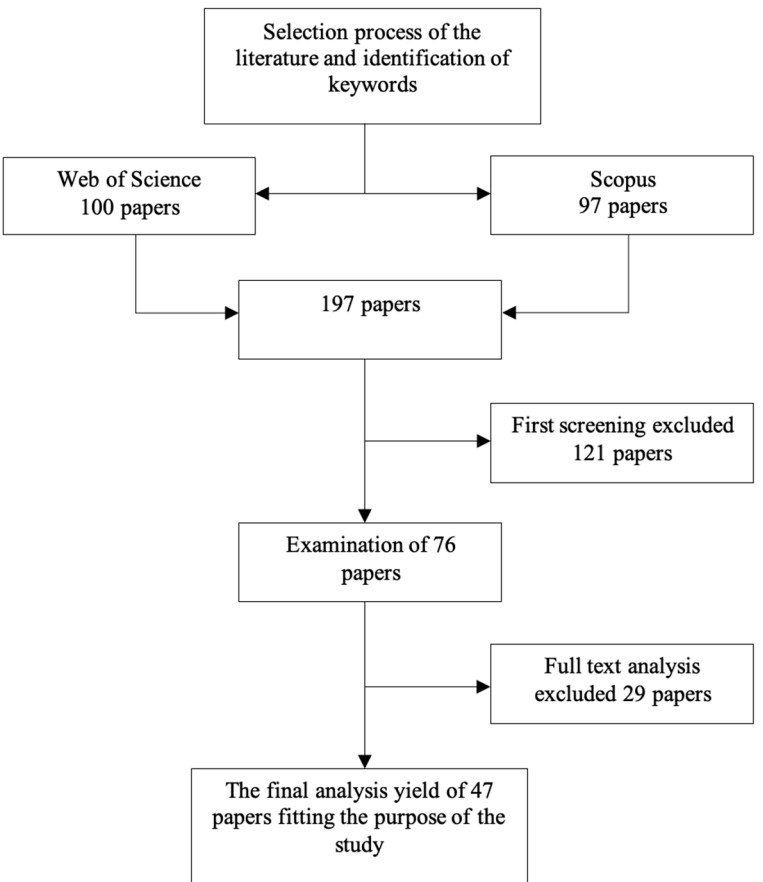

**Figure 2.** The research process flowchart.

Figure 3 also shows that there has been an increased interest in TM, recruitment strategies, and banking-and financial sector-related discussions since 2007, with the exception of 2009 and 2012, when the number of published papers fell compared to the previous years. We also observed considerable fluctuation in the number of published papers from 2018 to 2020.

Table 2 illustrates which journals have published the most articles on TM, recruitment tactics, and the financial aspect, as well as the methodology used for such research. Interestingly, only two journals had more than one published paper on the topic, the *International Journal of Economic Research,* which published 2 papers or 4% of the total number of papers, and the *Management of Science Letters*, which also published 2 papers or 4% of the total number of papers.

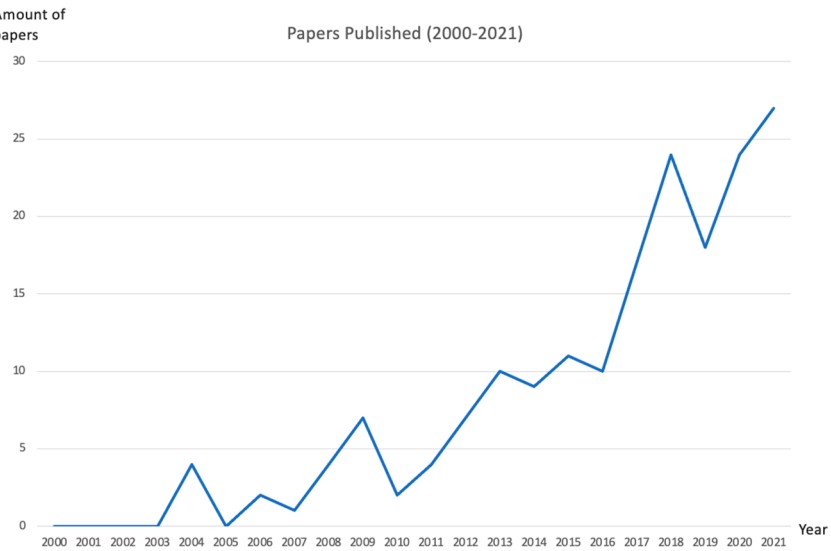

**Figure 3.** The number of papers published.

**Table 2.** Journals publishing papers on the chosen topics and the research methods employed.

| Journal | Number of Publications | Percentage | Methodology | | | |
|---|---|---|---|---|---|---|
| | | | Qualitative | Quantitative | Multiple | Other |
| International Journal of Economic Research | 2 | 4% | 0 | 1 | 1 | 0 |
| Management Science Letters | 2 | 4% | 0 | 2 | 0 | 0 |
| Other Publisher with one publication | 43 | 92% | 12 | 24 | 4 | 4 |
| Total | 47 | 100% | 12 | 27 | 5 | 4 |

Of the 47 selected papers, 43 were published in 43 different journals. Even though certain authors contributed to several published papers on the topic, the distribution of used publication outlets was notable.

In addition, Table 2 displays the methods used in the research. Qualitative approaches were used in 26% of the research (12 papers), whereas quantitative methods were the leading method of choice in 57% of the investigations (27 papers). Mixed methods were used in 10% of the studies (5 papers) and 8% (4 papers) applied other methods, such as, inter alia, observations and experiments.

Table 3 displays the complete list of the 47 selected papers. They are summarized by author, year of publication, name of the paper, name of the journal and keywords.

**Table 3.** Summary of selected papers.

| Author (Year) | Title | Journal | Keywords |
|---|---|---|---|
| Adeleye (2015) | Accelerating corporate transformation in emerging markets: The case of Firstbank | South Asian Journal of Business and Management Cases | Africa, Banking, Business leadership, Corporate Transformation, Organizational Renewal, Service leadership |
| Al-Azzam and Al-Qura'an (2018) | How knowledge management mediates the strategic role of talent management in enhancing customers' satisfaction | Pacific Business Review International | Talent Management, Strategies, Knowledge Management, Customer Satisfaction, Jordan Banking Sector |

**Table 3.** *Cont.*

| Author (Year) | Title | Journal | Keywords |
|---|---|---|---|
| Alnaimi and Rjoub (2021) | Perceived organizational support, psychological entitlement, and extra-role behavior: The mediating role of knowledge hiding behavior | Journal of Management & Organization | Perceived Organizational Support, Knowledge Hiding Behavior, Psychological Entitlement, Extra-role Behavior |
| Ali and Kasim (2019) | Talent management for Shariah auditors: Case study evidence from the practitioners | International Journal of Financial Research | Competency, Knowledge, Shariah Auditors, Skills, Talent Management |
| Al-Qura'an (2021) | Role of employer branding in enhancing the talent management strategies: applied study at commercial banks of Jordan | Independent Journal of Management & Production | Employer Branding, Talent Management, Commercial Banks, Jordan |
| Aman et al. (2018) | The impact of human resource management practices on innovative ability of employees moderated by organizational Culture | International Journal of Organizational Leadership | Human Resource Management, Innovative Ability, Organizational Culture |
| Chahal and Kumari (2013) | Examining talent management using CG as proxy measure: A case study of State Bank of India | Corporate Governance (Bingley) | Banking; Boardroom Performance; Business Performance; Corporate Governance; India |
| Charan (2011) | Banking on talent (talent management) | Development And Learning in Organizations | Career development, Employee development, High flyers, Succession planning |
| Chaudhry and Babin Dhas (2020) | Talent management practices in service sector: Evidences from literature review | International Journal of Pharmaceutical Research | Compensation Management, Competency Development, Employee Engagement, Human Capital, Talent Evaluation |
| Dang et al. (2020b) | The relationship between talent management and talented employees' performance: Empirical investigation in the Vietnamese banking sector | Management Science Letters | Banking Sector, Talent Management, Talented Employees' Performance, Vietnam |
| Dang et al. (2020a) | Talent conceptualization and talent management approaches in the Vietnamese Banking Sector | Journal of Asian Finance, Economics and Business | Banking Sector, Talent, Talent Conceptualization, Talent Management, Vietnam |
| Eliyana and Istyarini (2017) | The estimation and the fulfillment scenarios of human resources of sharia banking in Indonesia | Journal of Islamic Economics, Banking and Finance | Employee Engagement, Human Capital, Human Resource Management, Management Development, Performance Management |

**Table 3.** *Cont.*

| Author (Year) | Title | Journal | Keywords |
|---|---|---|---|
| Gaur et al. (2017) | Impact of frontline service employees' acculturation behaviors on customer satisfaction and commitment in intercultural service encounters | Journal of Service Theory and Practice | Acculturation, Assimilation, Frontline Employees, Integration, Intercultural Service Encounters, Marginalization, Separation |
| Glaister et al. (2021) | Talent management: managerial sense making in the wake of Omanization | International Journal of Human Resource Management | Institutional logics, Localization, Oman, Sense Making, Talent Management |
| Groysberg et al. (2008) | When "Stars" Migrate, Do They Still Perform Like Stars? | MIT Sloan Management Review | Leadership, Talent Management, Employee Performance, Sports, Team Building, Team Dynamics |
| Gulyani and Bhatnagar (2017) | Mediator analysis of passion for work in Indian millennials Relationship between protean career attitude and proactive work behavior | Career Development International | Behavior, Motivation (psychology), Employee behavior, Attitudes, Career development, Individual behavior |
| (Gulyani and Bhatnagar 2017) | The influence of visionary leadership, talent management, employee engagement, and employee motivation to job satisfaction and its implications for employee performance all divisions of bank bjb head office | International Journal of Scientific and Technology Research | Employee Engagement, Employee Performance, Job Satisfaction, SEM, Talent Management, Visionary Leadership, Work Motivation |
| Howe-Walsh (2015) | Bank stems the loss of employees returning from abroad: Talent-management system helps to keep people loyal | Human Resource Management International Digest | Banks, Multinationals, Repatriation, Retention, Talent Management |
| Hutt and Gopalakrishnan (2020) | Leadership humility and managing a multicultural workforce | South Asian Journal of Business Studies | Leadership, Organizational culture, Organizational identity, Organizational ambidexterity, South Asia |
| Ibrahim and AlOmari (2020) | The effect of talent management on innovation: Evidence from Jordanian banks | Management Science Letters | Banking Industry, Innovations; Jordan, Talent Management |
| Iqbal and Kamil (2017) | Talent management and succession planning on talent engagement at Islamic banks: The Malaysian bankers' perspectives | International Journal of Economic Research | Islamic banking, Succession planning, Talent engagement, Talent management |
| Kamil et al. (2018) | Talent development and retention from the bankers' perspectives: A study at Islamic Banks in Malaysia | Journal of Social Sciences Research | Islamic banks, Talent development, Talent management, Talent retention |

**Table 3.** *Cont.*

| Author (Year) | Title | Journal | Keywords |
|---|---|---|---|
| Khoram and Samadi (2013) | Relationship of talent management and organizational creativity in Maskan bank of Hamedan | Middle East Journal of Scientific Research | Hamedan, Maskan bank, Organizational Creativity, Talent management |
| Kokila and Ramalingam (2014) | Benchmarking HRM practices among banking sectors in Chennai | International Journal of Economic Research | Benchmarking, Human resource, Recruitment, Talent Management, Training |
| Lukman and Kee (2020) | Talent retention and job performance: The mediating role of perceived organizational support | Journal of Critical Reviews | Motivation, Talent Attraction. Talent Management, Talent Retention, Training |
| Maheshwari et al. (2017) | Exploring HR practitioners' perspective on employer branding and its role in organisational attractiveness and talent management | International Journal of Organizational Analysis | Banking, Employer Branding, Human Resource Management, Mauritius, Organisational Attractiveness, Talent Management |
| Mangion-Thornley (2021) | Coaching in the context of talent management: An ambivalent practice | International Journal of Evidence Based Coaching and Mentoring | Banking Sector, Coaching, Ethics, Social Exchange Theory, Talent Management |
| Mensah (2019a) | Talent management and employee outcomes: A psychological contract fulfilment perspective | Public Organization Review | Affective Commitment, Ghana, OrganisationalCitizenship Behaviours, Psychological-Contract Fulfilment, Talent Management |
| Mensah (2019b) | Talent management and talented employees' attitudes: mediating role of perceived organisational support | International Review of Administrative Sciences | Banking Sector, Commitment, Perceived Organisational Support, Quit Intention, Satisfaction, Talent Management |
| Mensah et al. (2016) | Unlocking the "black box" in the talent management employee performance relationship Evidence from Ghana | Management Research Review | Banking Sector, Ghana, Organizational Citizenship Behaviours, Person-Organization Fit, Satisfaction, Talent Management |
| Mensah and Bawole (2018) | Testing the mediation effect of person-organisation fit on the relationship between talent management and talented employees' attitudes | International Journal of Manpower | Banking Sector, Ghana, Organizational Citizenship Behaviours, Person-Organization Fit; Satisfaction, Talent Management |
| Munir et al. (2016) | BCA's employer branding—The challenge ahead | Emerald Emerging Markets Case Studies 2016 | Banks/banking, Human Resource Management, Recruitment, Strategy |
| Napathorn (2020) | How do MNCs translate corporate talent management strategies into their subsidiaries? Evidence from MNCs in Thailand | Review of International Business and Strategy | Institutional Structures, Liability of Origin, MNCs From Developed Economies, MNCs From Emerging Economies, Qualitative research, Skill Shortage, Talent Management |

**Table 3.** *Cont.*

| Author (Year) | Title | Journal | Keywords |
|---|---|---|---|
| Noreen and Imran (2021) | Impact of talent management practices on financial performance: evidence from GCC banking sector | Middle East Journal of Management | Banking, Talent Management, Competence Training, Development, Financial Performance |
| Pranee et al. (2017) | A Thai banking industry organisational performance analysis | Journal for Global Business Advancement | AEC, ASEAN, Association Of Southeast Asian Nations, Association Of Southeast Nations Economic Community, Banking Industry, CFA, Confirmatory Factor Analysis, SEM, Service Quality, Structural Equation Modelling, Talent Management, Technological Innovation, Thailand |
| Ready et al. (2008) | Winning the race for talent in emerging markets | Harvard Business Review | N/A |
| Sabbagha et al. (2018) | Predicting staff retention from employee motivation and job satisfaction | Journal of Psychology In Africa | Employee Motivation, Employee Retention, Foreign Exchange, Job Satisfaction |
| Santoso et al. (2020) | Talent mapping: a strategic approach toward digitalization initiatives in the banking and financial technology (FinTech) industry in Indonesia | Journal of Science and Technology Policy Management | Banking Industry, Digitalization, FinTech, Human Resource Management, Industry 4.0, Talent mapping |
| Savaneviciene and Vilciauskaite (2017) | Practical application of exclusive and inclusive talent management strategy in companies | Business Management and Education | Talent, Talent Management, Human Resource Management, Exclusive Talent Management Strategy, Inclusive talent Management Strategy, Organisation |
| Sehatpour et al. (2021) | Talent management in government organizations: identification of challenges and ranking the solutions to address them | International Journal of Productivity and Performance Management | Government organization, MCDM, Talent, Talent management |
| Singh and Sabharwal (2021) | Talent Management: An Empirical Analysis of Its Antecedents and Consequences Applying Structural Equation Modeling | Purushartha | Creative leadership, Spiritual Development, Talent management, Vision |
| Strizhova (2017) | Work motivation measurements among financial sector employees consisting in the manager reserve and talent development programs | Organizatsion-naya Psikologiya | Work Motivation, Motivation, Talent Pool, Talent Management, Motivational Task, Motivational Map |

**Table 3.** *Cont.*

| Author (Year) | Title | Journal | Keywords |
|---|---|---|---|
| Tajuddin et al. (2015) | Using talent strategy as a hedging strategy to manage banking talent risks in Malaysia | International Business Management | Banking talent risks, Business Strategy, Global banking, Malaysia, Talent strategy |
| Thanh et al. (2021) | The conceptualization of talent and talent management within the banking sector in Southern Vietnam | IBIMA Business Review | Banking Sector, Talent Management, Vietnam |
| Wadhwa and Tripathi (2018) | Driving employee performance through talent management | International Journal of Environment, Workplace and Employment | Employee performance, Private Sector Banks, Public, Talent Management Practices |
| Wang (2018) | To relocate internationally or not to relocate internationally: a Taiwanese case study | Journal of Global Mobility–The Home of Expatriate Management Research | Expatriates, International assignments, Perceived Organizational Support For International Assignment, Perceived Value Of The International Assignment By The Organization, Perceived Value Of The International Assignment To One's Career |
| Yazdanshenas (2019) | Promoting human capital through talent management practices: Contextual role of psychological contracts | Kasetsart Journal of Social Sciences | Human Capital, Psychological Contracts, Skill Enhancement Practices, Talent Management |

*3.2. Research Focus by Regions*

The 47 publications were scrutinized based on the places on which the research was focused, i.e., nation, area, or what was labeled "world." This comprised investigations that were conducted using data originating from three or more separate continents. According to the study, 41 of the publications (87%) were focused on a certain country or area, while 6 studies (13%) addressed the issue from a global viewpoint. With 58% (28 papers), Asia received the most attention, and Africa was in second place with 17% (8 papers), followed by the United States and Europe, each with 4% (2 papers). Oceania and Russia were both at 2% (1 paper).

Most of the publications were extremely precise about the nation on which they focused, although a handful were not (see Figure 4). In particular, owing to their dominance, Asia and Africa have received the most attention. According to the report, a research gap based on geographical focus could be observed, with a specific paucity of studies concentrating on Europe and North America.

According to the study, the geographical effort was primarily focused on Asia, the Middle East, and Africa, which means that more research into Europe and the Nordic nations is required.

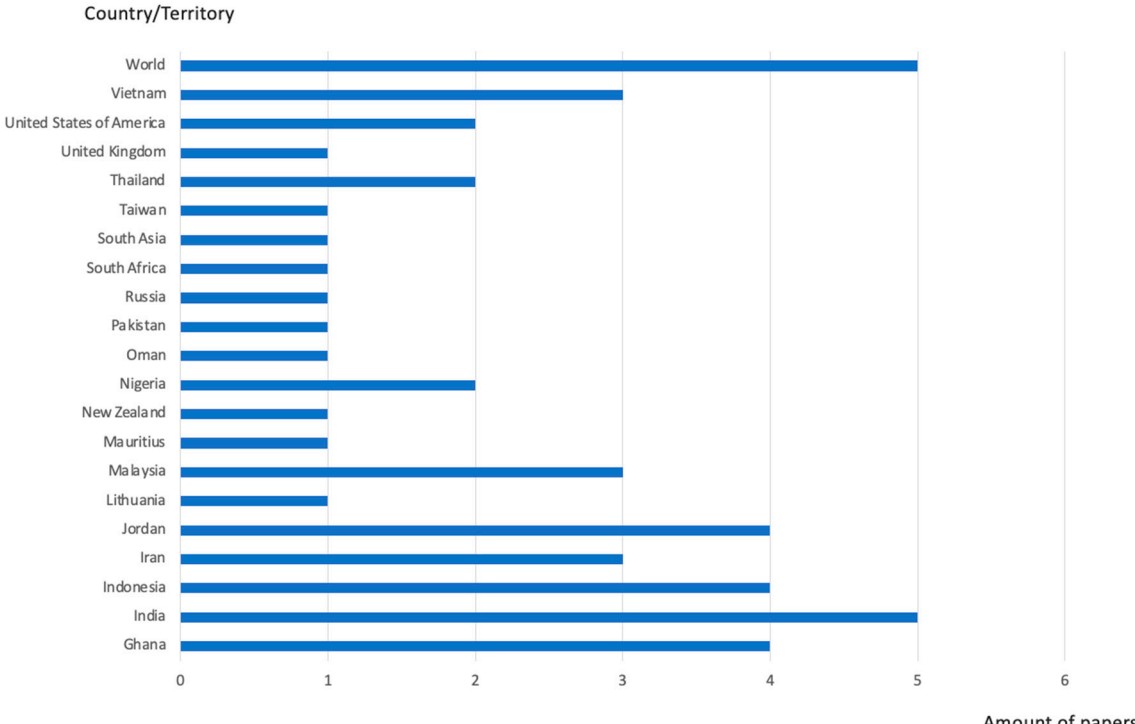

**Figure 4.** Research focus by geography.

### 3.3. Studies by a Theoretical Approach

In the literature, a lack of a theoretical approach could be observed. In 12 of the 47 studies, the theoretical approach was provided. Figure 5 demonstrates how research at the level of social conduct conforms to the social exchange theory, which was the most commonly used. According to the data, the social exchange theory accounted for 34% of the total number of papers (4 papers); various theories covered 25% (3 papers), the psychological contract theory covered 17% (2 papers); and the person-organization fit theory, the self-determination theory and the motivation theory each covered 8% (1 paper).

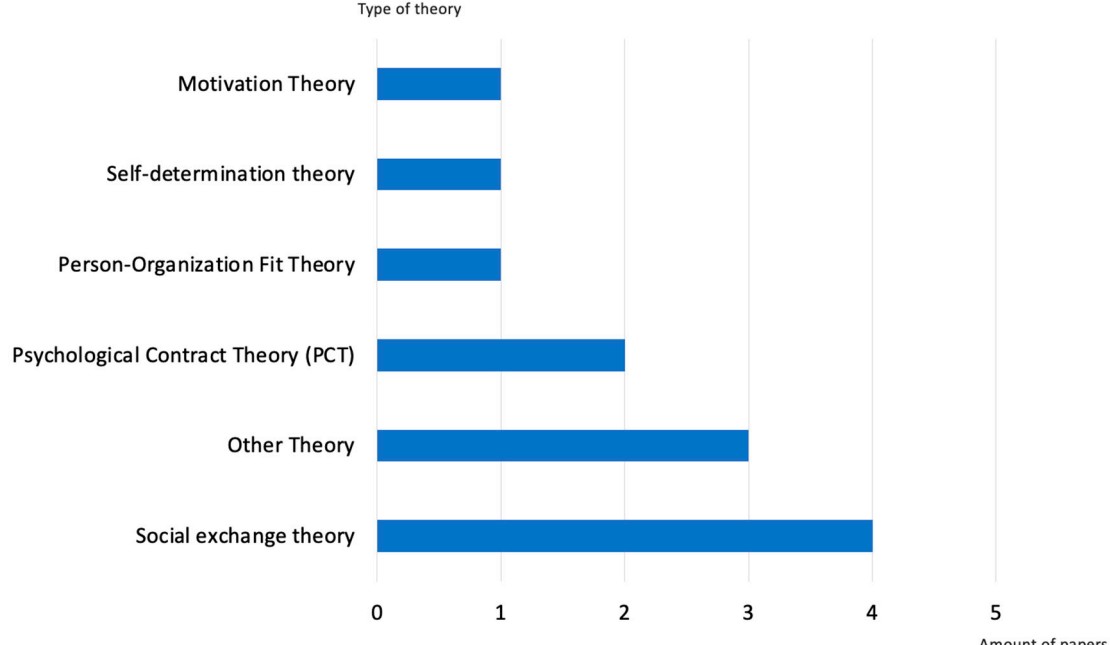

**Figure 5.** Overview of theoretical approaches.

The concepts investigated using the social exchange theory included how knowledge management mediates TM's strategic role in enhancing customer satisfaction, coaching in the context of TM, TM and talented employees' attitudes (the mediating role of perceived organizational support), and the unlocking of the "black box" in TM employee performance in the banking sector.

One paper investigated acculturation habits, while another explored localization policies (Omanization) via the use of TM, with a third examining the possible relationship between HR practice and benchmarking, as well as the link between total quality management (TQM) and HRM through the use of benchmarking.

### 3.4. Overview of Studies by Aim, Purpose, and Objective

The review reveals the aims, purposes, and objectives that were given by the authors of the 47 papers that were chosen for examination. Out of all the papers, 20% (9 publications; see Figure 6) sought to study how TM could motivate and influence employee performance in the financial sector.

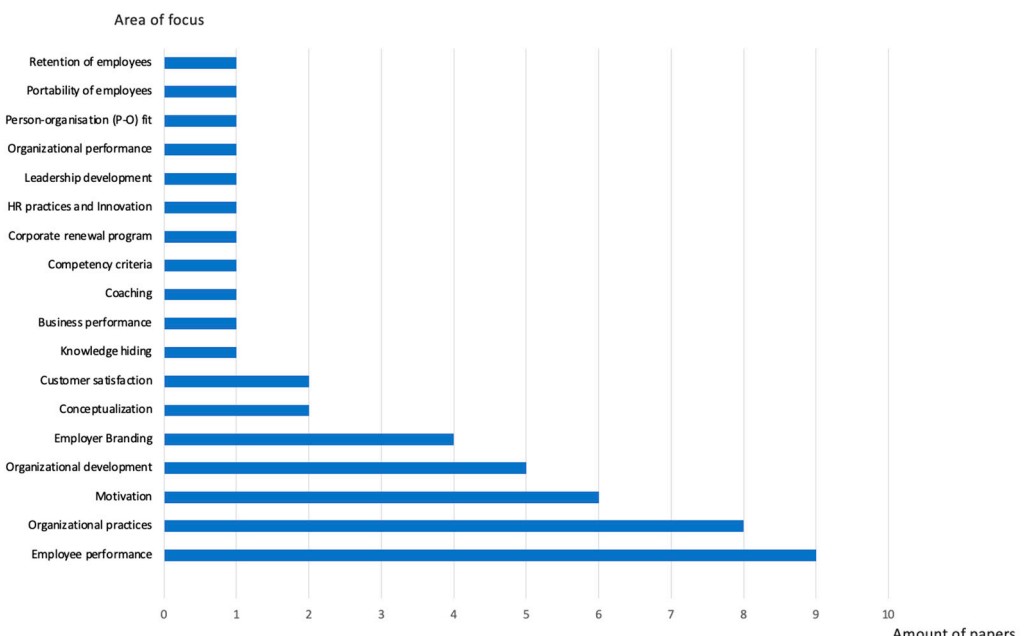

**Figure 6.** Results related to aims, purposes, and objectives of the studies.

In 17% of the studies (8 articles), researchers sought to analyze organizational practices, such as the examining of TM methods and their relevance, obstacles encountered in the TM process, and initiatives taken for TM in the service sector (Chaudhry and Babin Dhas 2020).

How managers utilize people management to make sense of localization policies (Omanization) and how leadership style may affect the formation of workplace culture among a global workforce to produce a collaborative, inventive, and high-performing firm are the areas of focus (Glaister et al. 2021; Hutt and Gopalakrishnan 2020).

Unsurprisingly, a focus on motivation was apparent in 14% of the reviewed publications (6 papers). In these cases, the researchers sought to investigate the relationship between protean career attitude (PCA) and proactive work behaviors (PWB), as well as to determine whether passion for work acts as a mediator for PCA and PWB (Gulyani and Bhatnagar 2017); the link between TM, product innovation, process innovation, and marketing innovation focusing on human capital (Ibrahim and AlOmari 2020); and the mechanism thereof (Mensah 2019b). The researchers were also interested in gaining a better understanding of the issues facing returning personnel, following an overseas assignment (Sabbagha et al. 2018).

Out of all the papers, 11% (5 of the reviewed publications) sought to explore organizational growth. This process included researching the link between TM and organizational innovation (Khoram and Samadi 2013), defining future talent skills based on existing conditions and future demands, and mapping talent in the banking and fintech industries (Santoso et al. 2020). The scholars also focused on ranking and prioritizing solutions to address the challenges for the successful implementation of TM in government banks (Sehatpour et al. 2021), as well as demonstrating that the use of talent strategies as a hedging strategies can manage banking talent risks (Tajuddin et al. 2015).

In 8% of the cases, the primary focus was on employer branding (4 of the analyzed papers). The authors of this case study wanted to know how much managerial leadership officials at the top, middle, and lower levels of practice understand about employer branding dimensions and TM strategies (Al-Qura'an 2021), what HR professionals and practitioners think about the role of employer branding in employer attractiveness and TM (Maheshwari et al. 2017), and how organizations can effectively increase employees' willingness to work (Wang 2018).

A total of 4% of the papers (2 papers) analyzed conceptualization, while 4% of the papers (2 papers) of the papers studied customer satisfaction. Other topics featured in 2% of the papers (1 paper) and focused on knowledge hiding, business performance, coaching, competency criteria, corporate renewal programs, HR practices and innovation, leadership development, organizational performance, person-organization (P-O) fit, portability of employees and retention of employees (see Figure 6).

### 3.5. Overviews and Frequency of Keywords

According to the analysis, 25 of the 47 examined publications incorporated industry classification in their keywords. A total of 7 articles (28%) classified the debate as the banking sector; 5 papers (20%) utilized the notion of banking; 3 papers (13%) used banking industry; and 2 papers (8%) used Islamic banking.

The following 8 papers (32%) each contributed with 1 paper each or 4% of the total number of papers. Their keywords were "banks," "commercial banks," "global banking," "Jordan banking sector," "Maskan bank," fin tech," "Industry 4.0," and "private sector banks" (see Table 4).

**Table 4.** Keywords used for describing the banking sector.

| Industry | Number of Studies | Percentage of Studies |
|---|---|---|
| Banking sector | 7 | 28% |
| Banking | 5 | 20% |
| Banking industry | 3 | 13% |
| Islamic banking | 2 | 8% |
| Banks | 1 | 4% |
| Commercial banks | 1 | 4% |
| Global banking | 1 | 4% |
| Jordan banking sector | 1 | 4% |
| Maskan bank | 1 | 4% |
| Fin Tech | 1 | 4% |
| Industry 4.0 | 1 | 4% |
| Private sector banks | 1 | 4% |
| Total | 25 | 100% |

To assure employee performance, the general approach to TM in the banking sector highlights the need to reform or redesign leadership, TM, employee engagement, employee motivation, and work happiness, for example (Dang et al. 2020b; Eliyana and Istyarini 2017; Hayati 2020). Few studies have been conducted to investigate organizational strategies and how leadership style may affect the establishment of workplace culture among a global workforce when building a collaborative, inventive, and high-performing corporation

(Chaudhry and Babin Dhas 2020; Glaister et al. 2021). The discussion around Islamic banking within countries such as Malaysia, Jordan, and Iran was addressed in some papers. The focus of the discussion related to the impact of TM and succession planning on talent engagement in Islamic institutions (Iqbal and Kamil 2017; Kamil et al. 2018).

There was a focus on various topics and some of the discussions addressed employee engagement, such as visionary leadership, TM, employee engagement, employee motivation, job satisfaction, and employee performance, as well as the impact of visionary leadership on job satisfaction, the impact of TM on job satisfaction, the impact of employee engagement on job satisfaction, the impact of employee motivation on job satisfaction, and the impact of job satisfaction on employee performance (Hayati 2020). It should be emphasized that only 2 publications (1.4%) focused on career development and employer branding, respectively. For example, the approach used in the career development papers was to investigate the relationship between PCA and PWB and to determine whether passion for work acts as a mediator for PCA and PWB, using the theoretical underpinning of the self-determination theory (Gulyani and Bhatnagar 2017). Using employer branding as an example, the article sought to determine the extent to which managerial leaders at the top, middle, and lower levels of commercial banks put employer branding characteristics and TM strategies into effect (Al-Qura'an 2021).

Of the 47 chosen publications, 43 (91.5%) had more than one term referring to the study's topic. Table 5 shows the frequency of the keywords that researchers used to explain their themes, which were connected to TM and the banking sector. In the selected publications, the overall frequency of terms linked to explaining TM was 220. The most frequently used keyword was "talent management", which appeared in 29 papers (13%), followed by "banking sector", which appeared in 20 papers (9%). It should be noted that the authors have combined the following keywords "banking sector", "banking", "banks", "banking industry", "banks/banking", "commercial banks", "Jordan banking sector", "private sector banks", and "global banking". "Human resource management" appeared in 6 papers (3%), "countries" was used in 9 papers (4%); "employee engagement", "employee performance", "talent", "motivation", and "human capital" all respectively appeared in 3 papers (1.4%), as shown in Table 5. It should be recognized that the authors have limited the results in Table 5 to the 24 most frequently applied keywords. A full list can be observed in Appendix A.

**Table 5.** Most frequently used keywords.

| Frequency of Keywords | | | Frequency of Keywords | | |
|---|---|---|---|---|---|
| Keywords | | Percentage | Keywords | | Percentage |
| 1. Talent Management | 29 | 13% | 13. Work Motivation | 2 | 0.9% |
| 2. Banking Sector | 20 | 9% | 14. Islamic banking | 2 | 0.9% |
| 3. Human Resource Management | 6 | 3% | 15. Job satisfaction | 2 | 0.9% |
| 4. Employee Engagement | 3 | 1.4% | 16. Jordan | 2 | 0.9% |
| 5. Employee Performance | 3 | 1.4% | 17. Leadership | 2 | 0.9% |
| 6. Vietnam | 3 | 1.4% | 18. MNCs from developed economies | 2 | 0.9% |
| 7. Talent | 3 | 1.4% | 19. Organizational Culture | 2 | 0.9% |
| 8. Motivation | 3 | 1.4% | 20. Recruitment | 2 | 0.9% |
| 9. Human Capital | 3 | 1.4% | 21. Satisfaction | 2 | 0.9% |
| 10. Career development | 2 | 0.9% | 22. Succession planning | 2 | 0.9% |
| 11. Employer Branding | 2 | 0.9% | 23. Talent retention | 2 | 0.9% |
| 12. Ghana | 2 | 0.9% | 24. Training | 2 | 0.9% |
| | | | 25. Others | 117 | 53% |
| The total frequency of keywords | | | | 220 | 100% |

### 3.6. Analysis of Studies by Key Concepts

Table 6 shows how the important themes were used in the 47 publications that were chosen for the study. The concept "talent management" was mentioned the most fre-

quently in 31 of the evaluated publications evaluated (66%). The concept "human resource management", as well as other concepts, was used favorably in 8 studies (17%).

**Table 6.** Breakdown of the concepts found in the literature.

| Categories Related to the Concepts | Number of Studies | Percentage |
|---|---|---|
| Talent Management | 31 | 66% |
| HRM | 8 | 17% |
| Other | 8 | 17% |
| Total amounts of papers | 47 | 100% |

As shown in Appendix A, the selected papers are divided into five key topics and many subtopics. It is clear that many of the subtopics concern how TM can contribute to increased efficiency and a better work environment. The subtopics become quite specific and certainly expand in the direction of how the organization of the future must necessarily integrate TM into its efforts to succeed. The individual's competence localization and development are emphasized, which, together, make it necessary for the organizations to take a stand on how they intend to meet the challenges of the future on the employee front.

Subtopics concerning organizational culture address inter alia, the different layers that must be coordinated so that the objectives comply with the actual picture. Here, the management culture plays a significant role in relation to well-being and the working environment. A collaborative culture is necessary if an organization is to create a strong and appealing image that can attract and retain attractive employees.

Appendix A shows the important themes that academics link with TM in the banking sector. Furthermore, it demonstrates that research relating to TM in the business of banking is also classified as "human resource management", "organizational culture", "productivity", and "recruiting". In addition, many subtopics were investigated. Some examples were "talent management black box", "talent risk management", "competency development", "person-organization fit", "corporate renewal", "employee attraction", "corporate governance", "knowledge sharing", "localization policies (Omanization)", "job satisfaction", "affective commitment", "liability of origin", "talent evaluation", "technological innovation", "talent mapping" and "human capital". This knowledge is important when it comes to gaining knowledge about TM in the banking sector. When considering the selected papers, it is important to understand that there are many different preferences and focus areas; therefore, specificity in the definitions that one uses and in one's intentions is a requirement. TM can help to find many solutions in the banking sector, but it can also simply be a buzzword that is used in many contexts, but that organizations cannot make valuable. Overall, it is necessary to clarify what TM is and what it is not to be able to use it as a tool. The next step is to determine whether the TM opportunities match the challenges facing the banking sector and ultimately, whether the initiatives are financially sound.

## 4. Contributions and Suggestions for Future Research

The study synthesizes the contributions of existing publications and the authors' recommendations for future research on what remains unknown about TM in the banking sector (see Table 7).

Analyzing the contributions of the articles revealed that most of them aimed to gather practical information to guide the future direction of TM. Others attempted to fill an empirical vacuum and contribute to the literature with their study by structuring the knowledge of the performance of HRM in the banking sector. The articles explored the organizational culture- and performance-related challenges that the banking sector is addressing through recruiting, such as the relationship between TM practices and the performance of talented individuals. Additionally, the current research has gathered data on what drives banks to implement TM and suitable HR structures and how management tackles the challenges (Al-Azzam and Al-Qura'an 2018). Furthermore, the importance

of employer branding in strengthening people management strategy activities has been stressed (Aman et al. 2018). Moreover, the papers attempted to develop a framework for TM by observing industry actions regarding talent attractiveness and the impact of employer branding, as well as how the banking sector measures and reports its performance (Al-Qura'an 2021; Groysberg et al. 2008; Maheshwari et al. 2017).

**Table 7.** Suggestions for future studies.

| Subject | Suggestions for Future Studies |
| --- | --- |
| Organizational development | Execution of a corporate renewal program (Adeleye 2015)<br>Knowledge hiding in organizations (Alnaimi and Rjoub 2021)<br>The effect of talent management on innovation (Ibrahim and AlOmari 2020)<br>Relationship of talent management and organizational creativity (Khoram and Samadi 2013)<br>Improving our understanding of the mechanisms responsible for the relationship between TM and employee outcomes is important (Mensah 2019a)<br>Maintaining a corporate culture when applying talent management (Munir et al. 2016)<br>How do multinational corporations (MNCs) manage talented employees in other emerging economies (Napathorn 2020)<br>The effects of talent management (Pranee et al. 2017)<br>Mapping talent in the banking and FinTech industries (Santoso et al. 2020)<br>Promoting human capital through talent management practices (Yazdanshenas 2019) |
| Performance | The effect of talent management on customer satisfaction (Al-Azzam and Al-Qura'an 2018)<br>Systematic management of human capital (Ali and Kasim 2019)<br>When 'stars' migrate, do they still perform like stars? (Groysberg et al. 2008)<br>The relationship between protean career attitude (PCA) and proactive work behaviors (PWB) (Gulyani and Bhatnagar 2017)<br>The mechanism through which talent management affects talented employees' attitudes (Mensah 2019b)<br>Investigate the relationship between talent management (TM) practices and talented employees' performance (Mensah et al. 2016)<br>The impact of talent management practices on financial performance (Noreen and Imran 2021)<br>Practical application of exclusive and inclusive talent management (Savaneviciene and Vilciauskaite 2017)<br>Work motivation measurements (Strizhova 2017)<br>Driving employee performance through talent management (Wadhwa and Tripathi 2018) |
| Motivation | The relationship between talent management (TM) practices and talented employees' performance (Dang et al. 2020b)<br>Talent-management system supports employee retention (Howe-Walsh 2015)<br>The effect leadership styles has on the workplace culture (Hutt and Gopalakrishnan 2020)<br>The relationship between talent development and talent retention (Kamil et al. 2018)<br>How does organizational support mediate the impact of talent retention (Lukman and Kee 2020)<br>coaching in the context of talent management (Mangion-Thornley 2021)<br>The impact job satisfaction and employee motivation has on retention (Sabbagha et al. 2018)<br>How can organizations effectively increase employees' willingness to relocate internationally (Wang 2018) |
| Strategy | The practice and impact of employer branding on talent management (Al-Qura'an 2021)<br>The lack of subordination, business ethics etc. In talent management (Chaudhry and Babin Dhas 2020)<br>Is talent management and Shariah a good fit? (Eliyana and Istyarini 2017)<br>Talent management as a tool to localize (Glaister et al. 2021)<br>Talent management and succession planning (Iqbal and Kamil 2017)<br>Benchmarking as a tool for improved talent management (Kokila and Ramalingam 2014)<br>Employer branding for service organizations' image and attraction as an employer (Maheshwari et al. 2017)<br>The effect of person-organization fit in talent management (Mensah and Bawole 2018)<br>Adjusting talent management strategies from home market to new markets (Ready et al. 2008)<br>Talent management in government organizations (Sehatpour et al. 2021)<br>Talent strategy as a hedging strategy to manage banking talent risks (Tajuddin et al. 2015) |

As the researchers state, practical implications include information about the organizations that have implemented TM practices and operate across borders, something that requires the organizations to adopt and adapt to cultural differences so that their actions appeal to stakeholders, in both local and worldwide markets (Hutt and Gopalakrishnan

2020; Chaudhry and Babin Dhas 2020; Dang et al. 2020b). Scholars have focused on the application of TM in Islamic banking and how Sharia auditors assimilate TM in organizational cultures (Ali and Kasim 2019; Eliyana and Istyarini 2017; Iqbal and Kamil 2017; Kamil et al. 2018).

Banks must demonstrate to stakeholders that they have met their TM objectives, mission, and strategy (Garavan et al. 2012; Ready et al. 2014). However, Sparrow (2019) claims that TM methods are defective and are preventing managers from making better judgments. In these sorts of companies, where TM procedures are anticipated to play a critical role in their operations, such as performance and motivation, there is a dearth of information about TM (including strategic) (Krzywdzinski 2019; Weisblat 2019). In 2012, Cascio and Boudreau stated the following: "Leaders at all levels now understand the vital importance of having talented employees who are motivated and aligned with the organization's strategy but relatively few have a deep understanding of how to be systematic in planning for and achieving this important condition for sustained success" (Cascio and Boudreau 2012). For a long time, managers were primarily concerned with a bank's economic and financial success short-term, ignoring the long-term effects (Pasichnyk 2014). Because of the developments indicated above in the banking sector over the last few decades, the authors believe it is critical to understand the state of the art for TM in a specific context—the banking sector. As a result, the purpose of this study is to present an overview of the primary studies on TM in the banking sector via a systematic literature review, as well as some suggestions for future research. As far as the authors are aware, this is the first systematic literature review to address this topic in this context.

The long-term economic advantages of establishing a sustainable recruitment strategy that includes TM techniques and attitudes towards progressive HRM have been studied. Furthermore, attention has been paid to the influence that service leadership can have on the future direction of recruiting and corporate performance (Groysberg et al. 2008; Hayati 2020; Hutt and Gopalakrishnan 2020; Ready et al. 2008).

With regard to prospective future studies and certain unsolved issues (Al-Azzam and Al-Qura'an 2018), the nature and effect of TM methods on customer satisfaction in the banking sector, as well as the mediating influence of knowledge management on this connection, were noted. Al-Azzam and Al-Qura'an (2018) proposed that TM strategies have a positive relationship with knowledge management, which in turn has a positive influence on customer satisfaction, and that further research should be conducted to investigate the impact of four types of acculturation behaviors of frontline service employees (assimilation, separation, integration, and marginalization) on customer satisfaction and customer commitment (Gaur et al. 2017). Adeleye (2015) highlighted the need to further investigate how established firms in a dynamic market can successfully implement a corporate renewal program in a hypercompetitive business environment, suggesting that quantitative research will provide newer insights than their study of the topic.

The goals of future study will include determining the link between HR practices and innovation in the banking sector, as well as the moderating impacts of organizational culture on this relationship (Aman et al. 2018). Furthermore, the influence of TM on business performance has been examined in a public sector bank by assessing TM using corporate governance as a proxy measure and examining its impact on business success (Chahal and Kumari 2013). It is necessary to determine, assess, and investigate visionary leadership, talent management, employee engagement, employee motivation, work satisfaction, and employee performance, as these factors are focal for organizational progress. Furthermore, the impact of visionary leadership on job satisfaction, the impact of TM on job satisfaction, the impact of employee engagement on job satisfaction, the impact of employee motivation on job satisfaction, and the impact of job satisfaction on employee performance have all been discussed (Hayati 2020). Further suggestions for future studies are displayed in Table 7.

## 5. Discussion

The goal of this study was to consider TM as it is used in the financial sector and specifically in the banking sector to understand what is already known about the subject and where research gaps can be observed.

This paper is thorough, since it covers studies over a long period of time (21 years) from the year 2000 to 2021 and encompasses as much issue-relevant research as feasibly possible for this time period. The investigation reveals how TM is linked to the banking sector through both primary topics and subtopics. The main topics, such as TM and HRM, are discussed and defined in a variety of ways. For example, the term TM is diffuse in the literature, since its definitions are dissimilar. The varying definition can lead to castings that can take many forms, and this is the case with TM, where there is an incredible amount of subtopics and, thus many possibilities. The positive aspect of TM being agile means that there is an indication of a broad interest, but it is negative when it means limited intensity. The study also highlights TM as one of the most significant variables for lucrative and efficient banking operations, the development of new banking products, and the delivery of superior services to consumers (Haines and St-Onge 2012; Vemic-Djurkovic et al. 2013). Our recommendation is that this attentiveness is relevant to the industry, as employee attraction, care, and retention are some of the crucial elements for companies in knowledge-intensive industries. In addition to a considerable cost when securing the "right" employees, a crucial factor for a company is to acquire a competitive advantage in a hostile market. The way in which an organization identifies, develops and redeploys personnel possessing a large amount of talent in a structured manner leads to a stronger competitive position within the market. This process is carried out by implementing what is called an inclusive TM strategy that allows all employees to receive recognition for having talent. An inclusive TM strategy demands a constant assessment and dedication from the employer to ensure that every employee is in a position that most suits their individual talents.

Conversely, companies can apply exclusive TM strategies. Using the argument that high-potential employees, ceteris paribus, add more value to the organization than employees who are considered average (Aguinis and O'Boyle 2014), it makes sense that high-potential employees' commitment and motivation becomes a priority when prioritizing and allocating the HR budget, assuming that the assets pay off. To ensure that the strategy is implemented in a reasonable manner, the organization must operate with transparency during the implementation process. A lack of transparency could have a negative effect on other employees, as such employees could end up with false expectations, and if their expectations are not met, these employees could become demotivated. Many HR professionals and business leaders have witnessed the influence of employing people who are mediocre simply to fill open positions, which became clear in the years of the global economic growth (2002–2007), and an understanding of the significance of hiring the "right" people has now become a priority for many companies (Dewhurst et al. 2012).

Our findings' proposals for future TM study propose a deeper exploration of the knowledge hidden in companies (Alnaimi and Rjoub 2021), maintaining a corporate culture when applying TM (Munir et al. 2016), and identifying and recruiting talent in the banking and fintech industries (Santoso et al. 2020). Other issues include the effect of TM on customer satisfaction (Al-Azzam and Al-Qura'an 2018), the ways through which TM influences the attitudes of brilliant workers (Mensah 2019b), how TM techniques affect financial performance (Noreen and Imran 2021), and the systematic management of human capital (Ali and Kasim 2019). The wide scope of the literature illustrates the complexity associated with TM. Many directions and many layers can be explored and understood in order to take advantage of existing opportunities and manage the risks that are to be found. Both academics and practitioners will be required to invest in these issues in order to find sustainable solutions. It must also be remembered that TM is a field that is constantly in motion, as its requirements and needs are constantly changing, which means that all stakeholders must understand the importance of building an agile organization, which would means increased re-costs and, in turn, returns in the future. With an emphasis on

sustainability, Bourdeau and Ramstad (2005) suggested that businesses must go beyond the traditional paradigm of pleasing stakeholders. Employees are a stakeholder group; thus, this transition is inevitable. Organizations must incorporate the goals of social sustainability goals into their overall strategic aims. According to Bourdeau and Ramstad (2005), sustainability "includes such aims as social responsibility, support for employees and other people's rights, diversity, nature preservation, and economic contribution." According to the literature, HR and TM are rarely confronted with the matter of sustainability, which must change.

Furthermore, there must be a greater focus on how TM systems can support employee retention (Howe-Walsh 2015; Kamil et al. 2018; Lukman and Kee 2020), increases in an organization's employees' willingness to relocate internationally (Wang 2018; Napathorn 2020; Tharenou 2008), provide coaching in TM (Mangion-Thornley 2021; Taconis 2018; Al Aina and Atan 2020), and reveal the impact of job satisfaction and employee motivation has on retention (Sabbagha et al. 2018; Alsubaie and Isouard 2019; Liu et al. 2022). Additionally, future research could address the impact of employer branding practices on TM (Al-Qura'an 2021; Maheshwari et al. 2017; Munir et al. 2016), whether or not TM and Sharia are a suitable fit (Eliyana and Istyarini 2017; Iqbal and Kamil 2017; Ali and Kasim 2019), whether TM can be a tool for an organization to localize (Glaister et al. 2021; Hartmann et al. 2010; Al Ariss 2014), and the lack of business ethics in TM (Chaudhry and Babin Dhas 2020; Dries 2013; Kwon and Jang 2021).

It is fair to conclude that the field is approaching adolescence and that it has a long, winding path ahead of it before reaching maturity. So far, the path has been good in terms of expanding our understanding of TM and how organizations in the banking sector are coping with the inherent obstacles. However, there is a need for care because it appears to be employed in a very loose way on many instances, which may have detrimental ramifications for conceptual and theoretical growth, which is the basis of building a critical research area. Furthermore, researchers must push intellectual limits and work on making this a viable field of research that contributes not just to academic understanding and theory but also helps bridge the often-mentioned practice divide. If the area is to mature, the fairly fragmented form of the literature must be brought closer together towards a more shared paradigm, and it is this that we encourage researchers to give the greatest attention. To do so most successfully, higher degrees of interdisciplinary and multidisciplinary research are likely to be required, which are often discussed in academic circles, yet appear to be rather uncommon. Not only does there need to be improvement from a conceptual and theoretical standpoint, but there also needs to be considerably enhanced empirical research that may tremendously aid in resolving the aforementioned fragmentation. While the assessment refuted prior claims that this was a subject dominated by conceptual work, there is no space for complacency in terms of the quality and amount of primary research. It is necessary to hear from HR managers, the top management team, middle and line managers, consultants, recruiting agencies, employees, and their representatives. Furthermore, there is a need to shift towards more generalizable research. There is a lot of room for quantitative study in this field, but one major problem is the absence of limits in the literature. It would be extremely impossible to conduct research with high levels of reliability and validity without the establishment of constructs.

In terms of a theoretical framework, the authors propose an investigation to examine the application of established theories, such as the motivation theory and the social exchange theory (Mangion-Thornley 2021). This investigation examines the relevant mechanism and perceived organizational support through which TM practices may affect the organization.

## 6. Conclusions

Academics are becoming increasingly interested in TM in the banking sector. This growing interest in TM seems to be related to the significant shortage of talented employees who specifically possess required competencies. Although various scholars have discussed

TM and the banking sector, most of the literature on the topic has been published in popular literature, rather than in the scientific peer-reviewed literature. According to the study, Asia, the Middle East and Africa were the focus of most of the research, which means that further research that concerns Europe and the Nordic countries is required.

This study seeks to address the issue of how TM is implemented in the banking sector and the financial sector as a whole, using a systematic literature review to determine what is already known about the topic. The primary contributions of this study are to provide a broad review of academic publications on TM, HRM, and the banking sector, which are accomplished by mapping the existing knowledge of how the industry conducts its practices and how the industry intends to respond to future demand for talent. This study adds to the literature by identifying key TM themes, as well as research gaps and future study prospects, and expanding on current knowledge of TM in the business of banking. This study suggests that TM studies emphasize how managerial qualifications are the groundwork for an organization's ability to attract and retain highly skilled employees. The focus must be on the correct leadership style and managerial qualities, as well as the organizational support that is provided to employees in the banking sector. As part of banks' desire for competitive advantages in relation to the market, a mapping of the TM strategy, as well as its compliance with the company's overall strategy, is important. The links between a transparent TM practice and a wholehearted follow-up provide the company with the opportunity to develop new ways of thinking, both in terms of organization and in terms of products and services, which contribute to potential organizational creativity and innovation. The link is also established via the mechanisms that are responsible for the relationship between TM and employee outcomes.

The field of talent management has a fragmented body of knowledge, and concepts of talent and talent management remain vague (Sparrow 2019). This study investigates two perspectives on the talent management discussion. The first question is whether the current literature should lead us to conclude that the field has become fragmented and requires streamlining, or whether there is a pattern and a process of increasing coherence that can be detected in the various directions that have emerged. Therefore, the purpose of the study is both to investigate how much literature there is on the subject but also to explore whether the field is moving in a certain direction.

The connection between organizational culture and performance is important. Talented employees with a long academic education or specialized knowledge are motivated by other factors, which differ from those of the employees who have a different profile. For example, talented employees in banks' market departments require a competitive culture where financial goals are clear and rewarded. Here, the academics are focused on a balance of work tasks, leisure, salary, the employer's image, and personal development. The organization must understand and act on this balance if it wishes to attract and retain employees within this segment, which would ultimately lead to better performance. This evaluation limited itself to scholarly publications from two databases and used inclusion and exclusion criteria. The selection was confined to English-language publications that concentrated on TM, HRM, and the business of banking, which may have resulted in the exclusion of relevant papers that were written in other languages.

**Author Contributions:** Conceptualization, U.T.; methodology, U.T.; writing—original draft preparation, U.T.; writing—review and editing, U.T., T.G. and S.G.; visualization, U.T.; supervision, T.G. and S.G.; project administration, U.T.; All authors have read and agreed to the published version of the manuscript.

**Funding:** This research received no external funding.

**Institutional Review Board Statement:** Not applicable.

**Informed Consent Statement:** Not applicable.

**Data Availability Statement:** Not applicable.

**Conflicts of Interest:** The authors declare no conflict of interest.

## Appendix A

**Table A1.** Key topics and subtopics of talent management in the banking sector.

| Key Topic | Sub-Topic |
|---|---|
| Talent Management | Effects of talent management, technological innovation in talent management, talent development programs, talent management practices in the service sector, relationship between talent management and talented employees' performance, conceptualization of talent, talent outcome, talent management black box, talent management in government organizations, talent risk management, conceptualization, talent, competency development, exclusive talent management strategy, high flyers, inclusive talent management strategy, psychological entitlement, talent conceptualization, talent engagement, talent management practices |
| Human Resource Management | International relocation of employees, international recruitment, competency criteria, portability of exceptional employees, talent development, talent retention, coaching, employees attitudes, person-organization fit, career development, succession planning, training, competence training, creative leadership, development, employee development, human resource, knowledge, knowledge management, marginalization, organization, person-organization fit, quit intention, separation, skill enhancement practices, skills, spiritual development, visionary leadership |
| Organizational Culture | Corporate renewal, acculturation behaviors on customer satisfaction, commitment in intercultural service encounters, leadership style, impact the development of workplace culture, collaborative culture, innovative culture, high-performing organization, visionary leadership, employer branding, Social exchange, employee attraction, organizational creativity, talent management strategy diffusion, leadership development, satisfaction, assimilation, behavior, boardroom performance, business leadership, business strategy, compensation management, corporate governance, corporate transformation, expatriates, digitalization, ethics, FinTech, government organization, Industry 4.0, innovations, innovative ability, institutional structures, localization, management development, organizational citizenship behaviors, organizational ambidexterity, organizational identity, organizational renewal, public, repatriation, sense making, social exchange theory, strategies, strategy, talent development, talent strategy, team building, team dynamics, vision, |
| Productivity | Competition in hypercompetitive market, knowledge sharing, knowledge hiding, motivation, service quality, effect of talent management strategies on customer satisfaction, localization policies (omanization) through their use of talent management, employee engagement, job satisfaction, employee performance, ,returning expatriate employees, the link between talent management product innovation process innovation and marketing innovation, financial performance, work motivation, affective commitment, benchmarking, business performance, commitment, competency, employee behavior, extra-role behavior, employee motivation, financial performance, foreign exchange, frontline employees, individual behavior, institutional logics, intercultural service encounters, international assignments, liability of origin, motivational map, motivational task, performance management, service leadership, service quality, talent evaluation, talented employees' performance, technological innovation |
| Recruitment | Recruitment strategy, staff retention, using talent strategy as a hedging strategy to manage banking talent risks, talent engagement, talent mapping, human capital, employee retention, integration, organizational attractiveness, psychological contracts, psychological-contract fulfilment, skill shortage, talent attraction, talent pool |

**Appendix B**

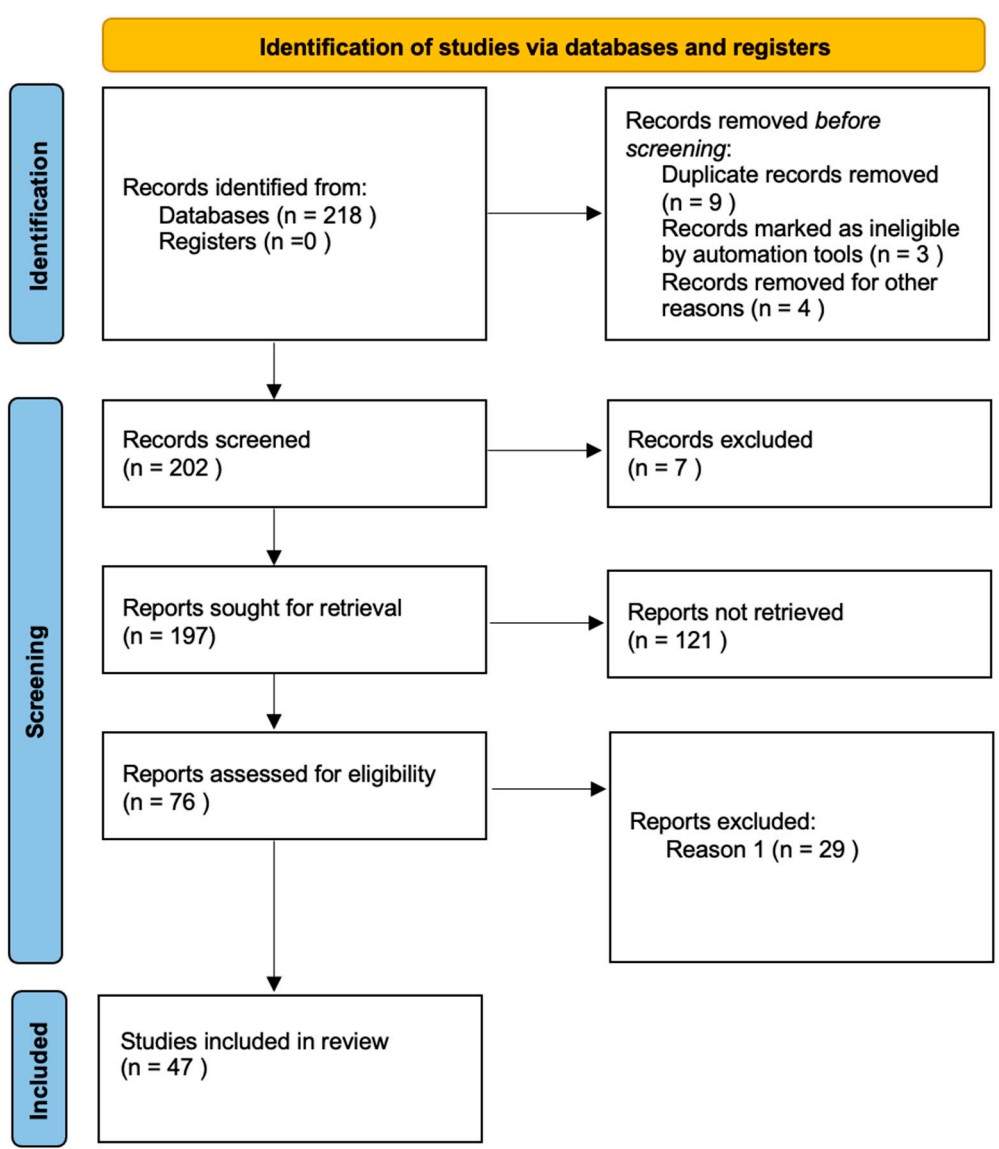

**Figure A1.** PRISMA flow diagram (Page et al. 2021).

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
