# Peer review of "Talent Management in the Banking Sector: A Systematic Literature Review"

_admsci, doi:10.3390/admsci12020061_

Round 1

Reviewer 1 Report

A very well written paper, solid methodology and results.

Minor comments, questions.

Row 41-why “managerial” credentials? What about others? Wording?

Row 69-recommendations for future research-from the review? Yours? Both?

Rows 72-83 – the language is not clear about the time frame you refer to- if this is about the early years, make the luanguge clear; if not, and you refer to present time-you have to update the references-they are old.

Row 119-check APA reference to use of first name…

Rows 175-8 and 186-8; when “…” you must use page no. per APA; same 243-245

Row 298-red in full- by who?

Rows 334-5-diffusion-is this the best word?

Rows 622-634; your recommendation? Or summary from your review. Be explicit using the right language. If it is yours then it is YOUR contribution.

Rows 675-697-those are the recommendations from your review, right?

Rows 698-711- your recommendation? Or summary from your review. Be explicit using the right language. If it is yours then it is YOUR contribution.

Author Response

Dear Reviewer,

Thank you for giving us the opportunity to submit a revised draft of my manuscript titled
“Talent management in the banking industry: A systematic literature review” to Administrative Sciences. We appreciate the time and effort that you have dedicated to providing your valuable feedback on my manuscript. We are also appreciative of your insightful comments on my paper.

We have been able to incorporate changes to reflect most of the suggestions provided.

In the attached Word document there is a point-by-point response to the reviewers’ comments and concerns.

Reviewer 2 Report

The article entitled: "Talent management in the banking industry: A systematic literature review" is interesting and well prepared. I recommend some suggestions and recommendations to improve the manuscript:

  • I recommend the authors to consider replacing the term (banking industry) with the banking sector,
  • Delete free line 64, I recommend changing the wording of lines 65-69.
  • Methods 3 Methods need to be introduced in Figure 1 of the text, authors must complete the reference to Figure 1. For example, line 242 or 247.
  • Delete free line 344.
  • I recommend the authors to add the name X-axis and Y-axis in Figure 3.
  • Table 3 needs to be aggregated or moved to the Apendix. I also recommend adding a synthesis of the findings from Table 3.
  • Figure 4 needs to be supplemented with X-axis and Y-axis and also Figure 5, Figure 6
  • Contributions and suggestions for future research. I recommend that the authors consider removing this chapter, it is not a standard (IMRAD). Personally, I would transfer the findings (contributions) to the section Discussion and Future research to Conclusions.
  • Lines 717-720: According to the study, Asia, the Middle East and Africa were the focus of most of the research, which means that further research that concerns Europe and the Nordic countries is required. Is the finding correct? Is this a conclusion from your article, or is further research really needed?
  • Line 741: The connection between culture and performance is important. Is the finding correct? I recommend adjusting the wording, what culture? National culture or organizational culture?
  • I recommend the authors to develop / add a part of the Limitations of the research (lines 750-752), because the study contains several limits.
  • I recommend the authors to see: (https://www.vosviewer.com/) and implement the results in their manuscript. My opinion is that the vosviewer application will help to improve (make) the submitted manuscript.
  • The last comment concerns the keyword (abbreviation) shrm (Sustainable Human Resource Management), which I think is absent throughout the manuscript. I recommend adding to the discussion or introduction to make it clear why the authors mention the keyword in question.

Final evaluation: Thank you to the authors for an interesting article, I recommend to consider my proposed recommendations. And I wish you good luck in your further research.

Author Response

(The authors gave the same response as above.)

Reviewer 3 Report

Introduction:
1. Justify the view that: The literature review presented will improve future TM research.
Contribution and suggestions:
1. Very vague suggestions. Provide specific directions for TM development.
2. Give grounds for drawing conclusions about whether only the number of publications or something more?
Discussion:
1. In the directions of TM development, provide publications of several authors pointing to the presented solution - not only the author / authors of one publication, e.g. in the field of job satisfaction and employee motivation, employer branding, business ethics.
2. A more detailed analysis of the content of the articles and drawing conclusions for the future is necessary.

Author Response

(The authors gave the same response as above.)

Round 2

Reviewer 3 Report

None